# Generative Modeling of Individual Behavior at Scale

## Abstract

There has been a growing interest in using AI to model human behavior, particularly in domains where humans interact with this technology. While most existing work models human behavior at an aggregate level, our goal is to model behavior at the individual level. Recent approaches to *behavioral stylometry*—or the task of identifying a person from their actions alone—have shown promise in domains like chess, but these approaches are either not scalable (e.g., fine-tune a separate model for each person) or not generative, in that they cannot generate actions in the style of each person. We address these limitations by casting behavioral stylometry as a multi-task learning problem—where each *task* represents a distinct *person*—and use parameter-efficient fine-tuning (PEFT) methods to learn an explicit *style vector* for each person. Style vectors are generative: they selectively activate shared "skill" parameters to generate actions in the style of each person. They also induce a latent style space that we can interpret and manipulate algorithmically. In particular, we develop a general technique for *style steering* that identifies a subset of players with a desired style property, and steers a new player towards that property. We apply our approach to two very different games, at unprecedented scales: chess (47,864 players) and Rocket League (2,000 players).

## 1 Introduction

The rapid advances in machine learning in recent years has made it increasingly important to find constructive ways for humans to interact with this technology. Even in domains where AI has achieved proficiency or superhuman performance, it is often important to understand how humans approach these tasks. Such an understanding can help identify areas for improvement in humans, develop better AI partners or teachers, create more realistic or enjoyable experiences, and more. AI that solely aims to solve a task optimally often fails in these respects, because such solutions tend to be difficult to interpret, provide limited instructional value to humans, and can be awkward or unenjoyable to interact with.

A common method for capturing human behavior is behavioral cloning (BC), a form of imitation learning (Schaal, 1996) that applies supervised learning to fixed demonstrations collected for a given task. While traditionally used in domains such as robotics (Florence et al., 2022) and self-driving vehicles (Pomerleau, 1988), BC has seen increasing use in gaming, such as in Counter-Strike (Pearce & Zhu, 2022), Overcooked (Carroll et al., 2019), Minecraft (Schäfer et al., 2023), Bleeding Edge (Jelley et al., 2024), and chess (McIlroy-Young et al., 2020).

The above work focuses on modeling human behavior in aggregate, motivated by the goal of developing better AI partners, opponents, and training tools. However, we believe that the most value for such goals can be derived by modeling human behavior at the *individual* level, because this allows us to tailor solutions to the individual's needs (e.g., creating an AI training partner that targets an individual's weaknesses). To that end, recent work in chess has shown the most promise. McIlroy-Young et al. (2020) used behavior cloning to create a set of models called Maia that mimic human play at 9 different skill levels. By fine-tuning these models on the data of 400 individual players, they created 400 personalized models that achieve 4-5% higher move-matching (predicting the user's next move correctly) accuracy on average (McIlroy-Young et al., 2022b). The authors use these models to perform *behavioral stylometry* with high accuracy, where the goal is to identify which person played a given query set of games. In this case, they simply apply each of the 400 models to the query set and output the one with the highest accuracy. McIlroy-Young et al. (2021) propose a more scalable approach of training a Transformer-based embedding on the games of each

player, and use this to perform accurate stylometry across 2,844 players. In this case, they compute the embedding of the query set of games and match it to the closest player's embedding.

These approaches have different merits. The individualized approach creates a generative model for each player, but it is not scalable and shares only initial (base model) knowledge across the players; adding a new player requires fine-tuning a separate model. The embedding approach is much more scalable: it learns a compact (single-vector) representation of each player in a shared style space, and supports few-shot learning to embed a new player in this space. It cannot be used to generate moves, however, and hence cannot reason about player behavior in practice.

An ideal solution would combine these properties: generative, scalable, shared knowledge, compact representation. Our key insight for achieving this is to view behavioral stylometry as a multi-task learning problem, where each *task* represents an individual *person*. The goal here is to generalize across an initial set of players (tasks) while supporting few-shot learning of new players (tasks). To do this efficiently, we leverage recent advances in parameter-efficient fine-tuning (PEFT) (Ponti et al., 2023; Caccia et al., 2023). Specifically, we augment an existing BC model with a set of Low Rank Adapters (LoRAs) as well as a routing matrix that specifies a distribution over these adapters for each player. Unlike approaches that train a separate LoRA for each task, this modular design allows players to softly share parameters in a fine-grained manner. We apply this adapter framework to two very different gaming models (which we create): a modified version of the Maia model for chess, and a Transformer-based model for Rocket League, a 3D soccer video game played by cars in a caged arena. We chose Rocket League and chess because they have a large, public collection of human games that span a diversity of skill levels and playing styles, testing the scalability of our approach to tens of thousands of unique players.

Our methodology first trains a BC model to convergence across all player data; then, it fine-tunes the adapters and routing matrix on per-player data. The base models we train outperform the state-of-the-art BC models for Rocket League and chess. Our fine-tuning process encourages the adapters to learn different *latent skills* that explain the variance between players, while each row of the routing matrix induces a weight distribution over these skills. We call each row the *style vector* for the corresponding player. Style vectors are versatile and powerful. They support few-shot learning which enables stylometry at scale. They induce a generative model for each player that we can run and observe. They induce a shared style space that we can interpret and manipulate algorithmically. Leveraging these properties, we develop a general, human-interpretable technique for *style steering* that identifies a subset of players who exhibit a desired style property, and steers a new player towards that property.

This paper makes the following contributions:

1. We develop a methodology for applying PEFT techniques to model individual human behavior, and create style vectors for behaviors. Style vectors capture a wide diversity of playing styles and strengths; they can be combined, interpolated, and steered, while reflecting consistent changes to playing style and strength.
2. We perform behavioral stylometry at an unprecedented scale for chess (47,864 players, 94.4% accuracy) and Rocket League (2,000 players, 86.7% accuracy), using a query set of 100 games. Our per-player generative models achieve move-matching accuracy in the range 45-69% for chess and 44-72% for Rocket League, even for players with very few (e.g., 50) games.
3. We present several analyses and applications of style vectors, showing that the black-box PEFT adapters are interpretable and editable. We include two examples of synthesizing new styles: interpolating weaker players to stronger ones, and steering player styles along human-interpretable properties.

## 2 BACKGROUND AND FRAMING

We frame behavioral stylometry and per-player generative modeling as a multitask learning problem, to which we apply PEFT methods. In multitask learning (Caruana, 1997; Ruder et al., 2019), we are given a collection of tasks $\mathcal{T} = (\mathcal{T}_1, \dots, \mathcal{T}_{|\mathcal{T}|})$, each task $\mathcal{T}_i$ associated with a dataset $\mathcal{D}_i = \{(x_1, y_1), ..., (x_{n_i}, y_{n_i})\}$. Multitask learning exploits the similarities among related training tasks by transferring knowledge among them; ideally, this builds representations that are easily adaptable to new tasks using potentially few target examples. The premise of this paper is that modeling individual human behavior from a pool of players can be interpreted as a multitask learning problem.

In other words, each task $\mathcal{T}_i$ consists of modeling the behavior of a specific player $i$; and dataset $\mathcal{D}_i$ corresponds to the sequence of game actions taken by player $i$. Specifically, an $(x, y)$ tuple denotes a game state $x$ at a specific point in time during game, along with the action $y$ that player $i$ took in this state. For the rest of the paper, we use the notion of *tasks* and *players* interchangeably.

## 2.1 PARAMETER-EFFICIENT FINE-TUNING

Popularized in NLP, parameter-efficient fine-tuning (PEFT) (Houlsby et al., 2019; Hu et al., 2022; Liu et al., 2022) approaches have emerged as a scalable solution for adapting Large Language Models to several downstream tasks. Indeed, standard finetuning of pretrained LLMs requires updating (and storing) possibly billions of parameters for each task. PEFT methods instead freeze the pretrained model and inject a small set of trainable task-specific weights, or "adapters".

One such approach is the use of Low Rank Adapters (LoRA) (Hu et al., 2022), which modify linear transformations in the network by adding a learnable low rank shift

$$h = \left( \boldsymbol{W}_0 + \Delta \boldsymbol{W} \right) x = \left( \boldsymbol{W}_0 + \boldsymbol{A} \boldsymbol{B}^T \right) x. \tag{1}$$

Here, $\boldsymbol{W}_0 \in \mathbb{R}^{d \times d}$ are the (frozen) weights of the pre-trained model, and $\boldsymbol{A}, \boldsymbol{B} \in \mathbb{R}^{d \times r}$ the learnable low-rank parameters of rank $r \ll d$. With this approach, practitioners can trade off parameter efficiency with expressivity by increasing the rank $r$ of the transformation.

## 2.2 POLYTROPON AND MULTI-HEAD ADAPTER ROUTING

Standard PEFT methods such as LoRA can adapt a pretrained model for a given task. In multitask settings, training a separate set of adapters for each task is suboptimal, as it does not enable any sharing of information, or *transfer*, across similar tasks. On the other hand, using the same set of adapters for all tasks risks *negative interference* (Wang et al., 2021) across dissimilar tasks , which may harm optimization and performance. Polytropon (Ponti et al., 2019) (`Poly`) addresses this transfer/interference tradeoff by softly sharing parameters across tasks. That is, each `Poly` layer contains 1) an inventory of LoRA adapters

$$\mathcal{M} = \{ \boldsymbol{A}^{(1)} \boldsymbol{B}^{(1)}, \, \dots \, , \, \boldsymbol{A}^{(m)} \boldsymbol{B}^{(m)} \},$$

with $m \ll |\mathcal{T}|$, and 2) a task-routing matrix $\boldsymbol{Z} \in \mathbb{R}^{|\mathcal{T}| \times m}$, where $\boldsymbol{Z}_\tau \in \mathbb{R}^m$ specifies task $\tau$'s distribution over the shared modules. This formulation allows similar tasks to share adapters, while allowing dissimilar tasks to have non-overlapping parameters. The collection of adapters $\mathcal{M}$ can be interpreted as capturing different facets of knowledge, or *latent skills*, of the full multitask distribution.

At each forward pass, `Poly` LoRA adapters for task $\tau$ are constructed as follows:

$$\boldsymbol{A}^\tau = \sum_i \alpha_i \boldsymbol{A}^{(i)}; \; \boldsymbol{B}^\tau = \sum_i \alpha_i \boldsymbol{B}^{(i)} \tag{Poly}$$

where $\alpha_i = \texttt{softmax}(\boldsymbol{Z}\,[\tau])_i$ denotes the mixing weight of the $i$-th adapter in the inventory, and $\boldsymbol{A}^{(i)}, \boldsymbol{B}^{(i)}, \boldsymbol{A}^\tau, \boldsymbol{B}^\tau \in \mathbb{R}^{d \times r}$. Here, the $\tau$-th row of the routing matrix $\boldsymbol{Z}$ is effectively selecting which adapter modules to include in the linear combination. In our setting, where each task consists of modeling an individual, $\boldsymbol{Z}\,[\tau]$ specifies which latent skills are activated for user $\tau$; we call this their *style vector*. As per Eqn 1, the final output of the linear mapping modified with a `Poly` LoRA adapter becomes $h = \left( \boldsymbol{W}_0 + \boldsymbol{A}^\tau (\boldsymbol{B}^\tau)^T \right) x$.

In `Poly`, the module combination step remains *coarse*, as only linear combinations of the existing modules can be generated. Caccia et al. (2023) propose a more fine-grained module combination approach, called Multi-Head Routing (`MHR`), which is what we use in our work. Similar to Multi-Head Attention (Vaswani et al., 2017), the input dimension of $\boldsymbol{A}$ (and output dimensions of $\boldsymbol{B}$) are partitioned into $h$ heads, where a `Poly`-style procedure occurs for each head. The resulting parameters from each head are then concatenated, recovering the full input (and output) dimensions. See A.1 for more details.

**Routing-only fine-tuning.** While LoRA adapters can reduce the parameter cost from billions to millions (Liu et al., 2022), training the adapters for each new task can still be prohibitive when dealing with thousands of tasks. To this end, Caccia et al. (2023) proposed routing-only finetuning, where after an initial phase of pretraining, the adapter modules are fixed, and only the routing parameters $\boldsymbol{Z}$ are learned for a new task. This reduces the parameter cost for each additional task by several orders of magnitude, while maintaining similar performance. We use this method for few-shot learning.

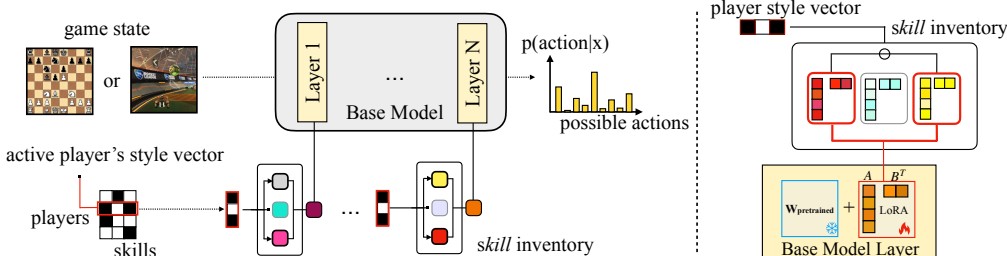

Figure 1: (left) Our overall architecture. We augment a base model with a set of MHR adapters and a routing matrix composed of each player's style vector. (right) Detailed view of an MHR layer, showing a skill inventory of adapters shared across players. The player's style vector specifies which skills are active (in this case, the first and third) to generate the final low-rank weight shift that is applied to the (frozen) base model layer.

## 3 ML METHODOLOGY

In this section, we detail our methodology for creating a generative model of individual behavior that enables our style analyses. Our methodology applies to any behavior cloning scenario with access to human demonstrations from multiple individuals. To demonstrate this generality, we apply it to two very different games: chess and Rocket League. We start with a base model for each and apply the MHR adapter framework to it, and then discuss model training and evaluation.

### 3.1 MODEL ARCHITECTURE

For chess, we follow McIlroy-Young et al. (2022b) and use the Squeeze-and-Excitation (S&E) Residual Network (Hu et al., 2018) as a base model, but with a deeper and wider configuration (see A.3). At every residual block, an additional 2-layer MLP rescales the residual output along the channel dimension to explicitly model channel interdependencies. The input is a 112-channel $8\times8$ image representation of the chess board; the output is the predicted move encoded as a 1858-dimensional one-hot vector. The total parameters is 15.7M. For Rocket League, we use the GPT-2 architecture from Radford et al. (2019) with a dimensionality of 768, 12 attention heads, and 12 layers. The input is a 49-dimensional vector with game physics information; the output is 8 heads: 5 with 3 bins of [-1, 0, 1] and 3 binary heads for a total of 1944 possible action combinations. The model has no embedding layer, as the game data points are passed directly as tokens after processing. The total parameters is 87.7M.

To enable user-based adaptation, we incorporate the MHR adapters described in §2.2 into our base models, as illustrated in Fig. 1. In chess, for every linear transformation in the MLP used for channel-wise rescaling, we add an MHR layer built of LoRA adapters with rank 16, for a total of $12\times2 = 24$ MHR layers. We use an adapter inventory of size 32 and a multi-head routing strategy with 8 heads. Therefore, for each user we must learn $32\times8 = 256$ routing parameters as their style vector. This yields 5M additional parameters. For Rocket League, we attach the adapters to the fully connected layer of each transformer block, resulting in 12 MHR layers of LoRAs with rank 16. We use an inventory size of 16 and 64 heads. This yields 13.8M additional parameters. To facilitate interpretability and style analysis, we use the same routing (style vector) across all MHR layers.

### 3.2 DATA COLLECTION AND PARTITIONING

We use data from the largest open-source online chess platform, Lichess.org (Duplessis, 2021), which boasts a database of over 4.8 billion games. We collected Blitz games played between 2013 and 2020 inclusive—these are games with 3 or 5 minutes per side, optionally with a few seconds of time increment per move—and applied the same player filtering criteria as McIlroy-Young et al. (2022b). The resulting dataset comprises 47,864 unique players and over 244 million games. (See A.3 for a discussion on data imbalance.) For Rocket League, we collect data from a large open-source replay database, Ballchasing.com (CantFlyRL, 2024). We use 2.2 million 1v1 replays from 2015 to mid-2022, totalling several decades of human game play hours at 5 minutes per game. After parsing, each Rocket League game state is a vector holding the player's 3D position, linear and angular velocity, boost remaining, rotation, and team; we also include the opponent's state and the position, linear and angular velocity of the ball. Given a game state, we have to predict the user's throttle, steer (while grounded), pitch, yaw, roll (while aerial), jump, boost, and handbrake.

Additional logic was needed to correct for missing aerial controls and inconsistent sampling rates (24-27hz). We describe our full data processing procedure, and the challenges we faced, in A.4.

We divide the set of players into a few subsets to support our training methodology. The *base player* set comprises all data and is used to train the base models. The *fine-tuning player* set is used to fine-tune the `MHR` architecture shown in Fig. 1. (For both, we split each player's data into 80/10/10 for train/test/validation.) The *few-shot player* set is used for few-shot learning based on a reference set of 100 games per player. For our chess experiments, to enable a direct comparison with prior work, we create an additional fine-tuning player set consisting of the same 400 players used in those studies. Currently, we treat each player's data holistically, but in principle one could partition a player in different ways to perform a finer analysis of their playing style. We explore this in A.5.

## 3.3 MODEL TRAINING AND EVALUATION

**Base model.** We train our base Maia (McIlroy-Young et al., 2020) model for chess using data from a base player set of all 47,864 players, treating this as a classification task of predicting human move $y$ made in chess position $x$, given a datapoint $(x, y)$. We use the same loss functions and evaluation criteria as the original Maia work: Maia's policy head uses a cross entropy loss while the value head uses MSE; the output of the policy head is used to evaluate the model's move-matching accuracy.

We train our Rocket League model using a base player set of over 800,000 players, though the vast majority of players have 5 games or fewer. We discretize the actions into 3 bins for throttle, steer, pitch, yaw, and roll, as most of this data is close to 0, -1, or 1. We use binary outputs for jump, boost, and handbrake. A next-move prediction is labelled correct if and only if all the outputs are correct.

**`MHR` fine-tuning.** To train the `MHR` LoRA adapters, we adopt the methodology used in Caccia et al. (2023): namely, we freeze the base model and fine-tune the `MHR` layers and routing matrix using data from a fine-tuning player set. Recall that the routing matrix $Z$ has a row (style vector) for each player in the fine-tuning set. Following Ponti et al. (2019), we use a two-speed learning rate, where the style vectors' learning rate is higher than the adapters', to enable better specialization.

For chess, we use two fine-tuning player sets in our experiments, creating two separate `MHR`-Maia models. The first set comprises all 47,864 players and is used to evaluate behavioral cloning and stylometry at very large scale. The second set is comprised of the same 400 players used by McIlroy-Young et al. (2022b), which we use to compare few-shot learning and stylometry results. For Rocket League, we train an `MHR`-Rocket model on a fine-tuning set of 2,000 players with 100 games each.

**Few-shot learning.** To perform few-shot learning on our `MHR` models, we perform the "routing-only fine-tuning" described in section 2.2 that additionally freezes all `MHR` LoRA adapters. Given a few-shot player, we add a (randomly-initialized) new row to $Z$ and fine-tune it on the player's reference set of games, eventually learning a style vector for the player. Using this style vector, we can invoke a generative model of the player and use it to evaluate move-matching accuracy, as described above. To perform stylometry, if the player is a *seen* player (i.e., part of the fine-tuning set), then a matching style vector already exists in $Z$, and we can find it using cosine similarity. Otherwise, if the player is *unseen*, then we simply repeat the few-shot learning process on a query set of games (from the same player), and compare this new style vector to the entries in $Z$. In general, the number of reference/query games required for few-shot learning is low (see Figure 9, A.3).

For chess, (unless stated otherwise), all of our few-shot experiments use the `MHR`-Maia model fine-tuned on the 400-player set from McIlroy-Young et al. (2022b). For Rocket League, the few-shot player set consists of 1,000 of the 2,000-player set used to fine-tune `MHR`-Rocket.

**Evaluation.** We evaluate a fine-tuned `MHR` model in two ways. First, we measure its move-matching accuracy, similar to how we evaluate the base models. However, since our `MHR` models provide a generative model for each player (conditioned on their style vector), we can separately evaluate each player's model by applying it to their test set and measuring move-matching accuracy. The overall move-matching accuracy for the model is the average of these per-player accuracies.

Our second evaluation method uses the model to perform behavioral stylometry among all players in the fine-tuning set. In theory, we could adopt the methodology of McIlroy-Young et al. (2022b) and compute the move-matching accuracy of every player applied to every other player's query set, but such a quadratic computation is infeasible beyond a few thousand players. Instead, we leverage our few-shot learning methodology above. That is, given a query set of games from some

player, we learn a new style vector in $Z$ for those games via few-shot learning, and compare this vector to every other vector in $Z$ using cosine similarity. We then output the player with the highest cosine similarity to the query set vector. In domains that focus on authenticating individuals (like biometrics), ROC curves and related metrics are used in place of top-1 classification. We show an ROC curve in Figure 11, where we treat any prediction other than the actual player as incorrect.

## 4 STYLE METHODOLOGY

The style vectors in $Z$ represent distinct distributions over latent skills that give us a starting point for comparing player styles. For example, our stylometry method above uses the cosine similarity between vectors to determine how similar or different players are. In addition, style vectors enable a much more powerful capability: the ability to synthesize new (human-like) styles.

To begin, we measure the intra-player consistency of style vectors by splitting a player's dataset into disjoint subsets of varying size, and use few-shot learning to learn a style vector for each subset. We then investigate inter-player consistency, by merging the datasets of two players and seeing if the style vector learned from the merged dataset is similar to simply taking the average of the two players' style vectors.

The latter method is notable because it actually creates a new playing style that is human-like, and yet has never been seen in the world. This suggests a more general approach to synthesizing new styles: interpolate between existing players using a convex combination of their style vectors. For example, we can smoothly transition from a weaker player's style to a stronger player's style. To determine the playing strength of a newly synthesized player, we can simulate games between them and the players they are derived from, by conditioning the `MHR` model on their respective style vectors. The results of these games yield a win rate, which can be converted to a strength rating.

Currently, our advanced style synthesis techniques focus on chess, where simulating games is cheap, evaluation heuristics are standardized, and a robust mapping exists between win rate and playing strength (the Elo rating system). Rocket League simulations are too costly at present and there are no standardized heuristics, but in principle the same methodology can be applied and we plan to make this practical in future work.

In order to make style comparisons more human interpretable, we again exploit the generative nature of our `MHR` models. Inspired by the concept probing technique used to analyze AlphaZero (a deep reinforcement learning chess engine) (McGrath et al., 2022), we use a set of human-coded heuristic functions found in Stockfish (a traditional chess engine) to evaluate a player's model. These functions capture concepts such as: king danger, bishop pair utilization, material imbalance, and so on. By invoking a player's model on a fixed set of chess positions, we can measure the change in the heuristic functions before and after their chosen move, and use this to summarize how much emphasis the player places on the corresponding human-interpretable concepts.

Finally, we combine the above methods to design a simple but general method for *steering* a player's style towards a specific, human-interpretable attribute $a$—such as king danger—while limiting the changes to other attributes (so as to preserve their style). We summarize this method in Algorithm 1. We first collect a set players $X$ who exhibit high values for attribute $a$—determined, for example, by running their generative models on a fixed set of game states. We then extract the common direction among these players, by averaging their style vectors and subtracting the population average. This yields a *style delta vector* that can be added to any player's style vector to elicit the desired change. We renormalize this vector to half of the L2 norm of the average norm of the vectors in our data. Since the heuristics used to generate these vectors are interpretable, humans can manually modify the behavior of a player's model to suit their needs.

## 5 EXPERIMENTS

In this section, we demonstrate two main findings. First, `MHR`-Maia performs competitively with prior methods for behavior cloning and stylometry in chess, and does so at an unprecedented scale. We also show that our approach can be applied to Rocket League, a challenging 3D video game environment, for both stylometry and move prediction. Second, we show that explicitly capturing style vectors allows us to analyze and manipulate the behavior of player models.

| Method | \|*Query*\| | \|*Universe*\| | \|*Query Games*\| | Random (%) | Acc. (%) |
|---|---|---|---|---|---|
| *Seen* few-shot players | | | | | |
| McIlroy-Young et al. (2022b) | 400 | 400 | 100 | 0.25 | 98.0 |
| McIlroy-Young et al. (2021) | 400 | 400 | 100 | 0.25 | 99.5 |
| MHR-Maia | 400 | 400 | 100 | 0.25 | **99.8** |
| McIlroy-Young et al. (2022b) | 400 | 400 | 30 | 0.25 | 94.0 |
| MHR-Maia | 400 | 400 | 30 | 0.25 | **98.8** |
| MHR-Maia | 10000 | 47864 | 100 | 0.002 | **94.4** |
| *Unseen* few-shot players | | | | | |
| McIlroy-Young et al. (2021) | 578 | 2844 | 100 | 0.035 | 79.1 |
| MHR-Maia (100 games) | 10000 | 10000 | 100 | 0.01 | **87.6** |

Table 1: Stylometry accuracy results. *Seen* few-shot players are a subset of the fine-tuning player set, unlike *unseen* players. Numbers for McIlroy-Young et al. (2022b) and McIlroy-Young et al. (2021) are borrowed from their respective papers.

## 5.1 BEHAVIORAL STYLOMETRY

For chess, we show that MHR-Maia perform competitively with previous behavioral stylometry methods for both seen and unseen players. Here, the goal is to predict which player produced a given set of games. We compare our approach to individual model fine-tuning (McIlroy-Young et al., 2022b), which fits a separate pre-trained Maia model to the data of each player, and to a Transformer-based embedding method (McIlroy-Young et al., 2021), which embeds players in a 512-dimensional style space based on their games. All reported accuracies are top-1 unless stated otherwise.

To perform stylometry on a query set of games, McIlroy-Young et al. (2022b) apply each player's fine-tuned Maia model on the query set and select the one with the highest move-matching accuracy. As seen in Table 1, this procedure works well, but it is very expensive—requiring a separate model for each player as well as computationally intensive inference calls on the entire query set per player.

In contrast, both the Transformer-based embedding and MHR-Maia scale to much larger numbers of players. The Transformer-based embedding needs only to embed the query games to compute a player vector, while MHR-Maia needs only to fit a new style vector on the query games. In either case, the produced vectors are compared to those in the player set to find the closest match, e.g., using cosine similarity. Table 1 compares both approaches, showing that MHR-Maia performs competitively or better, while scaling to a much larger universe size. When performing stylometry on players *seen* during MHR fine-tuning, we are able to achieve 94.4% stylometry accuracy given a universe of 47,864 seen players. To do so, we sample 10,000 query players from the set of seen players and fit a new style vector for each query player based on their 100-game query set. For stylometry on players *unseen* during MHR fine-tuning, we sample 10,000 new players form a held-out set and compute style vectors based on their 100-game reference sets, before fitting their style then repeat the above methodology above on their query set.

Although the Transformer-based embedding method can scale similar to our method, it is not a generative model (i.e., cannot play the game). Note that we omit the individual model fine-tuning method from the larger few-shot study due to its scalability limits. On an A100 80GB GPU, training individual models required roughly **20 A100-minutes per player** on average for Figure 2; thus, training on the full 47,864 player dataset would require thousands of A100-hours. In comparison, training MHR-Maia on the full dataset required roughly 7 A100 days, or around **12-13 A100-seconds per player**, an improvement of nearly two orders of magnitude. The inference costs were roughly equal, with MHR-Maia being marginally more expensive due to the added parameters.

For Rocket League, to the best of our knowledge, we are the first to attempt stylometry. We apply the same few-shot learning methodology to compute style vectors for 1,000 query players based on their 100-game reference sets, and then fit a new style vector for each query player based on their 100-game query set. (Recall that each game consists of 5 min of 1v1 gameplay.) For each of the 1,000 query players, our MHR-Rocket approach must correctly identify them among a universe of 2,000 players. We achieve an accuracy of **86.7%** (random performance being 0.05%), showcasing the validity of our approach even in a challenging 3D game scenario.

## 5.2 MOVE GENERATION

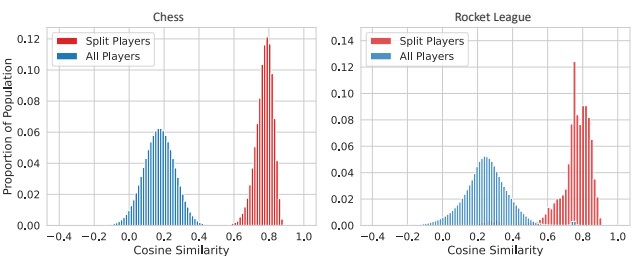
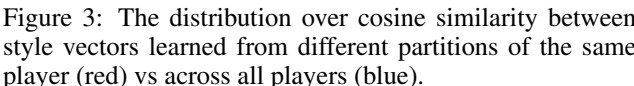
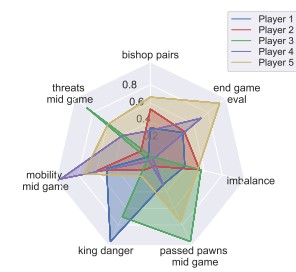

Figure 3: The distribution over cosine similarity between style vectors learned from different partitions of the same player (red) vs across all players (blue).

Figure 4: Comparing different player styles using human-interpretable evaluation metrics on a shared dataset.

A key feature of our MHR models is that they are generative, i.e., they can generate moves in the style of each individual player. This section evaluates the next move prediction accuracy of our models.

For chess, we compare the efficacy of our method to using individually fine-tuned models for each player. We do not compare to the Transformer-based embedding method because it is incapable of generating moves. Full fine-tuning of individual models generally results in superior performance compared to PEFT methods, as the increased parameter count produces more expressive models. However, their memory footprint is significantly bigger, making training and storage more challenging. That said, when it comes to modeling individual behavior in chess, MHR-Maia performs comparatively well despite using a much smaller parameter budget. Figure 2 shows that MHR-Maia matches individual model fine-tuning over a wide range of game counts. MHR-Maia has very competitive accuracy within 1% of individual fine-tuning on our improved version of the Maia model across all game counts. As described in the previous subsection, we achieve this with a compute cost that is practically a rounding error compared to fine-tuning individual models. Our results suggest that the model has already learned the set of shared skills required to differentiate between the players, so all that is needed from the few-shot learning is to find a proper recombination of the learned skills for each player, encoded in the new style vectors.

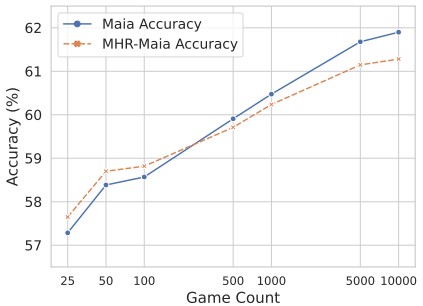

Figure 2: Accuracy at various game counts of the individual models (Maia) and our method (MHR-Maia). MHR-Maia is within 1% accuracy of individual models using a small fraction of the compute.

For Rocket League, we compare the next move prediction of our base model (trained on over 800,000 players) with MHR-Rocket (fine-tuned on 2,000 players), to validate that our user-based conditioning via style vectors generates better predictions. We find that MHR-Rocket increases the next move prediction accuracy from **53.1%** to **56.1%** (random performance being 0.05%), averaged over all 2,000 players. Moreover, the per-player move-matching accuracy ranges from 44-72%, suggesting that the model has learned a wide range of non-trivial behaviors.

## 5.3 ANALYSIS OF STYLE VECTORS

In this section, we explore the consistency of our style vectors within a player and across different players. We compare different model playing styles using human-interpretable metrics, and show how we generate new intermediate playing styles by averaging different players.

**Consistency within a single player.** To investigate if style vectors show consistency within a player, we first partition 50 players' datasets into disjoint subsets. Within a player, we use 50 splits for chess and 20 for Rocket League. The subsets are sampled across a wide range of dates, opposing players, and playing sessions. We then train a style vector on every split of every player, and compare these vectors using cosine similarity. We find that vectors corresponding to the same player are similar to each other, while having low similarity to other players and the general population. This is visualized in Figure 3. This suggests that our MHR models are able to associate distinct style characteristics with each player. We also find that these style characteristics are diverse: we sampled 5 random chess players and used their models to predict their preferred move across $2^{17}$ positions,

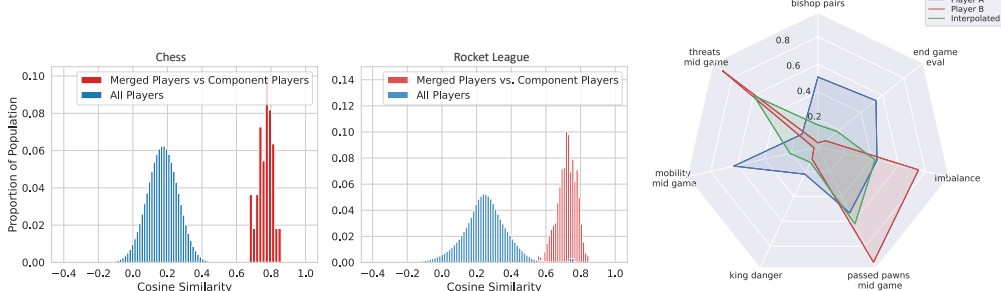

Figure 5: Cosine similarity between averaged style vectors of two players, and the learned style vectors on their merged datasets (red) vs across the full population (blue). The style of an intermediate play                                        ompone

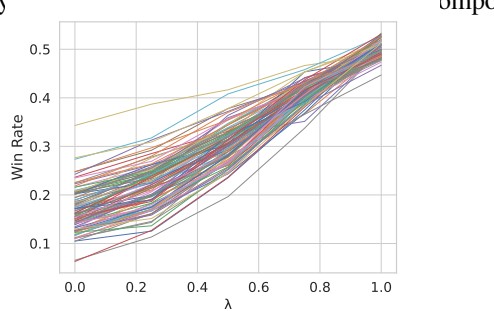

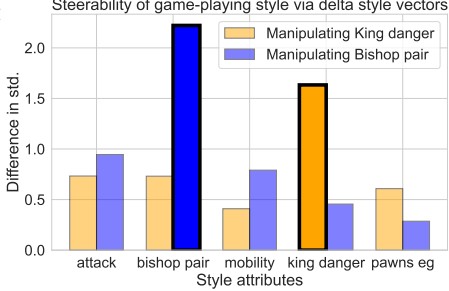

Figure 6: Win rate as randomly chosen weaker players are interpolated towards randomly chosen stronger players.

Figure 7: Increasing two Stockfish attributes (separately) for 2,000 random players using the style steering method from Section 5.4.

and then evaluated the move choices using a set of Stockfish heuristic functions as described in §4. Figure 4 shows the averaged metrics for each player, demonstrating that style vectors indeed capture a wide diversity of playing styles.

**Consistency across merged players.** To investigate if style vectors show consistency across different players, we consider the case of merging two players' datasets to create a new dataset representing the characteristics of both players. We train a style vector on the merged dataset, and then compare this to the vector obtained by simply averaging the style vectors of the component players. Figure 5 shows the results across a large population of players in Chess and Rocket League. As the figure shows, the style vectors trained on the merged datasets have high cosine similarity with the averaged vectors of the component players, while having low similarity with the general population. These results hold across Rocket League and chess. As a concrete example in chess, we took the averaged style vector of a random player pair, conditioned `MHR`-Maia on this vector to yield a generative model of the new player, and evaluated the player's move choices across 4096 games using the Stockfish heuristics mentioned in the previous section. The results are visualized in Figure 5, which shows that the style characteristics of the new player (green) intermediate between the styles of the component players (red, blue).

## 5.4 SYNTHESIS OF NEW STYLES

In this section, we investigate more advanced applications of style synthesis: interpolating between skill levels, and steering player styles along human-interpretable properties.

**Interpolating between players.** We show that interpolating between the style vectors of a weaker and stronger player results in new players whose skill levels also interpolates between the players. Here, we take 100 pairs of weak and strong player style vectors and gradually interpolate between them as $(1 - \lambda)u_w + \lambda u_s$, $0 \le \lambda \le 1$, where $u_w$ and $u_s$ are the respective vectors. For each value of $\lambda$, we simulate 1,000 games between the interpolated player and $u_s$, the stronger player. Figure 6 plots the win rate of the interpolated players as a function of $\lambda$ for each pair of players. This plot shows that the win rate increases in a roughly linear fashion as lambda increases, starting low and eventually winning roughly half the time, which is what we would expect from two players with the same style vectors.

**Steering player style.** We can directly control the playing style of a player using the steering method described in 4. Using the human-interpretable Stockfish heuristics, we identify players in our chess dataset with high ($> 2$ std) bishop pair utilization, and similarly players with high king danger. We use these players to compute style delta vectors corresponding to these attributes, and add them to 2,000 randomly sampled players' existing style vectors. Figure 7 shows the change in these players' Stockfish evaluations after adding the style delta vectors. Indeed, we see that the player's style is steered towards the attribute in question, with modest impact on other attributes. We show an additional application of this steering methodology in Appendix A.2, where we modify the outputs of an image generation model using style delta vectors.

## 6 Related Work

**Stylometry and player style modeling.** Originally referring to performing author attribution via statistical analysis of text (Tweedie et al., 1996; Neal et al., 2017), stylometry has since come to refer to the general task of identifying individuals given a set of samples or actions, and has found broad application for tasks such as handwriting recognition (Bromley et al., 1993), speaker verification (Wan et al., 2018), identifying programmers from code (Caliskan-Islam et al., 2015), determining user age and gender from blog posts (Goswami et al., 2009), and identifying characteristics of authors of scientific articles (Bergsma et al., 2012). In the context of gaming (covered in the introduction), stylometry is closely related to playstyle modeling, where the goal is to associate a player with a reference style, such as by building agents representative of different playstyles and find the closest behavioral match (Holmgård et al., 2014), or gathering gameplay data and applying methods such as clustering (Ingram et al., 2022), LDA (Gow et al., 2012), Bayesian approaches (Normoyle & Jensen, 2015), and sequential models (Valls-Vargas et al., 2015) to identify groups of players with similar styles. Kanervisto et al. (2021) characterizes an agent's behavior by analyzing the states that an agent sees (not actions). Unlike our work, these approaches either focus on aggregate play styles, or do not learn generative models of behavior that can be conditioned on an individual's style.

Our method for style synthesis is inspired by earlier work on vector arithmetic with embeddings (Church, 2017), as well as recent work on steering multiask models with task vectors (Ilharco et al., 2023). Our steering method is reminiscent of Radford et al. (2016), which manipulates the model's latent space to generate images containing specific attributes. Recently, Dravid et al. (2024) achieved similar results on images of people by training LoRAs and manipulating their weights.

**Parameter-efficient adaptation.** Approaches for efficient adaption of a pretrained model can be broadly grouped in two categories: injecting new parameters into a model, and soft-prompts. Houlsby et al. (2019) defines an adapter as a two-layer feed-forward neural network with a bottleneck representation, and are inserted before the multi-head attention layer in Transformers. Similar approaches have been used for cross-lingual transfer (Pfeiffer et al., 2020). Adapters have also been used in vision based multitask settings (Rebuffi et al., 2017). More recently, Ansell et al. (2022) propose to learn sparse masks, and show that these marks are composable, enabling zero-shot transfer. Lastly, Hu et al. (2022) learn low-rank shifts on the original weights, and (Liu et al., 2022) learns an elementwise multiplier of the pretrained model's activations. Adapters have also been used in multitask settings. Chronopoulou et al. (2023) independently trains adapters for each task, and merges parameters of relevant tasks to transfer to new ones. Another approach is the use of soft prompts (Lester et al., 2021), which appends learnable tokens to a natural language sequence. In a similar setting, Vu et al. (2021) learns a collection of soft-prompts from a multitask training set, and given a novel task, retrieves relevant prompts for efficient transfer.

## 7 Conclusion

We show that individual player behavior can be modeled at very large scale in games as different as chess and Rocket League. We cast this problem in the framework of multi-task learning and employ modular PEFT methods to learn a shared set of skills across players, modulated by a distinct style vector for each player. We use these style vectors to perform behavioral stylometry, analyze player styles, and synthesize and steer new styles. In future work, we would like to explore the use of these style vectors for fine-grained image editing, analysis of how humans respond to changes in task in the real world, and removal of dangerous or unwanted behaviors in language models.

## REPRODUCIBILITY AND ETHICS STATEMENTS

Our methodology sections and appendices describe our data collection, data processing, and model training and evaluation methodologies in detail. The data processing for Rocket League was particularly involved, as this required reverse-engineering certain features that are omitted from the replay files. Upon publication, we will have permission to upload our code and a subset of our processed data to a public repository. The (processed) Rocket League data will be newly contributed; the chess data will be similar to that of the Maia project (maia chess, 2024).

Our work creates generative models of individual behavior that can be used to impersonate or mimic real players. Although the data we train on is public, and the gaming environments (chess and Rocket League) relatively benign, the ability to mimic individuals efficiently at scale raises social and ethical questions related to the proliferation of such "mimetic models". The individuals targeted by these models, the deployment of these models (whether by the target individuals or third parties), and the interactions between others and these models, are all processes that need to be studied more deeply. A taxonomy of the parties and example scenarios were described in recent work by McIlroy-Young et al. (2022a).

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

## A APPENDIX

### A.1 MULTI-HEAD ADAPTER ROUTING

In `Poly`, the module combination step remains *coarse*, as only linear combinations of the existing modules can be generated. Caccia et al. (2023) propose a more fine-grained module combination approach, referred to as Multi-Head Routing (`MHR`). Similar to Multi-Head Attention (Vaswani et al., 2017), the input dimension of $A$ (and output dimensions of $B$) are partitioned into $h$ heads, where a `Poly`-style procedure occurs for each head. The resulting parameters from each head are then concatenated, recovering the full input (and output) dimensions. This makes the module combination step *piecewise linear*, with a separate task-routing matrix $Z$ learned for each head.

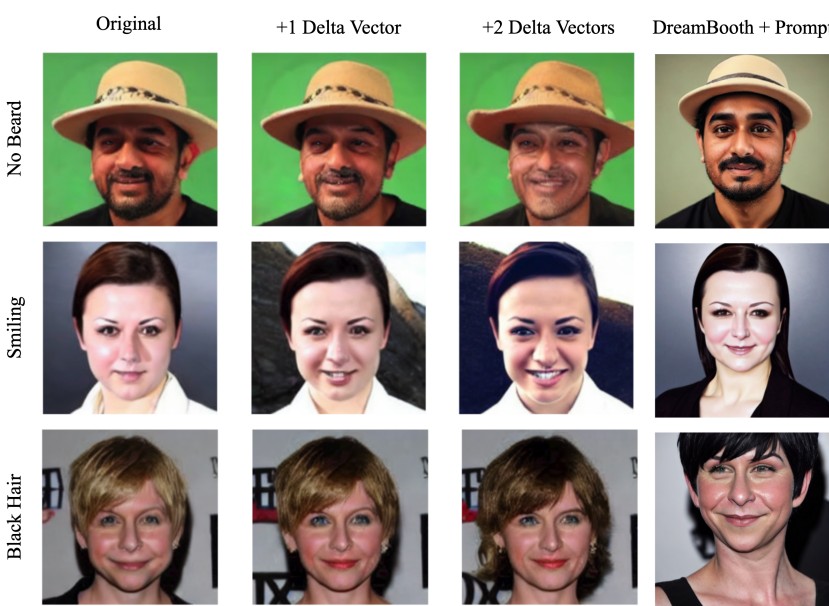

Figure 8: Images generated by steering Stable Diffusion 1.5 (Rombach et al., 2022) fine tuned with our method on the CelebA (Liu et al., 2015) dataset. We compare against using DreamBooth (Ruiz et al., 2023) on the original image and modifying the prompt.

Formally, a MHR layer learns a 3-dimensional task-routing tensor $\mathbf{Z} \in \mathbb{R}^{|\mathcal{T}| \times |\mathcal{M}| \times h}$. The 2D slice $\mathbf{Z}_{:,:,k} \in \mathbb{R}^{|\mathcal{T}| \times |\mathcal{M}|}$ of the tensor $\mathbf{Z}$ denotes the distribution over modules for the $k$-th head, and $\boldsymbol{W}[k] \in \mathbb{R}^{\frac{d}{h} \times r}$ the $k$-th partition along the rows of the matrix $\boldsymbol{W} \in \mathbb{R}^{d \times r}$. The adapter parameters $\boldsymbol{A}^\tau \in \mathbb{R}^{d \times r}$ for task $\tau$, and for each adapter layer, are computed as (similarly for $\boldsymbol{B}^\tau$):

$$\boldsymbol{A}_k^\tau = \sum_j \alpha_{i,k} \cdot \boldsymbol{A}_j[k] \ \text{ with } \ \boldsymbol{A}_k^\tau \in \mathbb{R}^{\frac{d}{h} \times r}, \tag{MHR}$$

$$\boldsymbol{A}^\tau = \texttt{concat}(\boldsymbol{A}_1^\tau, \dots, \boldsymbol{A}_h^\tau),$$

where $\alpha_{i,k} = \texttt{softmax}(\boldsymbol{Z}[\tau,:,k])_i$. Importantly, the number of LoRA adapter parameters does not increase with the number of heads. Only the task-routing parameters linearly increase with $h$ for MHR vs. Poly. However, this cost is negligible as the parameter count of the routing matrices is much smaller than for the LoRA modules themselves.

## A.2 STEERING DIFFUSION MODELS

To address questions about the generalizability of our method, we applied the exact style delta vector computation and steering algorithm outlined in Section A.6 to steer the outputs of an image generation diffusion model in a fine-grained manner. We use the CelebA Faces With Attributes dataset (Liu et al., 2015) to fine tune style vectors for 10,177 identities. We use Stable Diffusion 1.5 (Rombach et al., 2022) as our base model.

We compute "No Beard", "Smiling", and "Black Hair" style delta vectors using cosine similarities between the images and their respective CLIP (Radford et al., 2021) embeddings. Figure 8 shows sample images generated by applying these vectors, with the leftmost images being un-steered. We compare our results against using DreamBooth (Ruiz et al., 2023) with LoRA to fine-tune towards the original image, and adding "with no beard", "smiling", and "with black hair" to the prompt for the respective images.

Our method is able to achieve more granular control of the source image with minimal modifications to the style of the image. In contrast, while DreamBooth is able to change the specific feature we aim to steer, the remaining parts of the image are changed significantly.

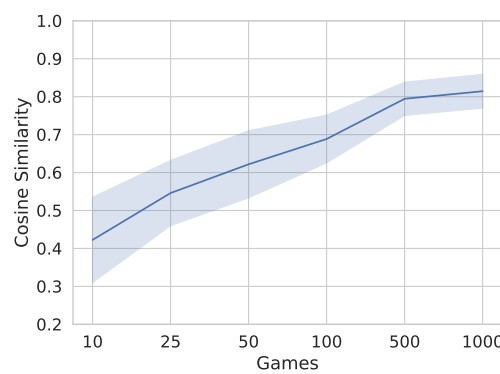

Figure 9: Cosine similarity of style vectors trained with varying game sizes compared to a style vector trained with 10,000 games, run on 50 players.

### A.3 MAIA ARCHITECTURE/DATA

Our base Maia architecture follows McIlroy-Young et al. (2022b) and uses the Squeeze-and-Excitation (S&E) Residual Network of (Hu et al., 2018). At every residual block, channel information is aggregated across spatial dimensions via a global pooling operation. The resulting vector is then processed by a 2-layer MLP, with a bottleneck representation compressing the number of channels by $r$. The output of this MLP is a one-dimensional vector used to scale the output of the residual block along the channel dimension. We use 12 residual blocks containing 256 filters, and a bottleneck compression factor of $r = 8$. We note that this differs from the base Maia model in McIlroy-Young et al. (2022b), which uses 64 filters and 6 residual blocks.

While our dataset has a median game count of 3,479 games, many players may have as few as 10-50 games, implying some degree of data imbalance. Our evaluation of few-shot learning shows that 100 games is sufficient to learn the style vector of an unseen player. However, one might still ask how accurately such a style vector is given a very small number of games. To explore this, we first split a player into disjoint sets of 10, 25, 50, 100, 500, and 1,000 games. We then train a style vector on each set. As a baseline, we train a style vector on 10,000 games and track the cosine similarity of the smaller-set style vectors relative to this baseline vector. We show the results in Figure 9.

### A.4 ROCKET LEAGUE ARCHITECTURE/DATA

The 1v1 replays dataset was scraped over the course of several weeks from the Ballchasing.com API using the Grand Champion subscription tier, though the API does have a slower free tier. This API yields raw game replays, which are uploaded by users either manually or using a community-made plugin for the game. The replays are in a binary format which must be parsed using community-made projects such as Carball (SaltieRL, 2024).

The Carball library allows us to convert the binary replay format to a more standard CSV format, which we save to a Cloud binary blob storage. The data present in both is a lossy reconstruction of game states, and requires some processing to be usable. In particular, the data is sampled at an inconsistent rate (varying between 24hz and 27hz), contains repeated physics ticks, and is missing action data for aerial controls (pitch, yaw, roll).

We resolve the issue of sampling rate and repeated ticks by removing repeated ticks, and doing a time-weighted resampling and interpolation to a standard 10hz for model training, though we found that 30hz also works well. Note that the actual game physics ticks occur at 120hz, so any value aligned with this should work. Without these changes, the model performs extremely poorly and is unable to navigate the arena.

We resolve the issue of missing aerial controls through the physics-based solver present in the Carball library. The estimation of these controls is not perfect, but it is sufficient for our purposes. Some previous community work has used inverse dynamics (Braaten, 2022) trained from rollouts of in-game bots to solve for these actions, though we opted to not use this due to the inconsistency in replay data sampling.

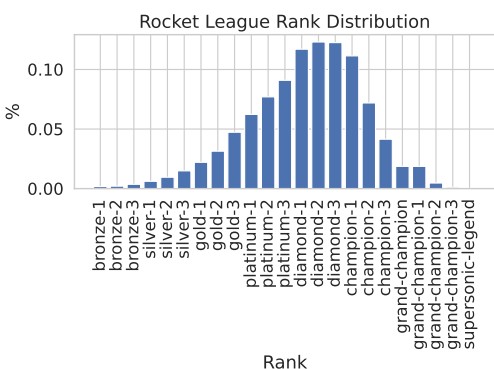

Figure 10: Skill distribution of Rocket League players in our dataset.

The data returned by the CSVs are fairly large, messy, and inconsistent. We apply the following transformations to the dataframe to bring the values closer to 0:

- Divide position by 2300

- Divide linear velocity by 23000

- Divide angular velocity by 5500

- Divide boost by 255

- Encode rotation Euler angles according to Zhou et al. (2020)

Additionally, when turning the data into tokens for use in our model, we add in an extra dimension to represent the team, and concatenate the opponent's data points along with the position, linear and angular velocity of the ball. We complete all of these transformations at runtime.

We also have to align the data returned by the simulators for Rocket League with the data used to train the model, RLBot (RLBot, 2017) and RLGym (Emery, 2021). Along with including an extra dimension to represent the team, we apply the following transformations to all samples obtained from the game:

- Divide position by 2300

- Divide linear velocity by 2300

- Divide angular velocity by 5.5

- Divide boost by 100

The skill distribution of the players in our dataset can be found in Figure 10.

### A.5 IMPLICIT STATIONARITY ASSUMPTIONS

Most of the existing work in chess assumes that a player remains stationary over time and across gameplay situations. However, in reality, a player's style may depend on the type of opponent they are facing, which opening is used, which stage of the game they are in (opening, middle, endgame), and so on. For instance, McIlroy-Young et al. (2021) observe that stylometry accuracy drops when removing the opening (e.g., the first 15 moves) moves, suggesting that the opening has an outsized effect on style identification. Our approach does not rely on these assumptions and can in principle be applied to arbitrary subsets of a player's data. For instance, one could split a player's data into opening, middlegame, and endgame moves and train a separate style vector for each. One could further split the data based on which defense the opponent uses, what time of the day it is, etc.. Despite treating players holistically and avoiding any splits of their data, we are still able to capture the peculiarities of each individual's playing style and perform stylometry with high accuracy. This also enables us to compare our results to those of prior work, which also treats player data holistically.

Figure 11: ROC Curve of Rocket League player detection.

## A.6 STYLE STEERING METHOD

---

**Algorithm 1** Style Delta Vector computation

---

**Input:**
$X$ : Style vectors of top-k players for attrib. $a$;
$P$ : Style vectors of all players in population
**Output** $\Delta_a$: Style delta vector for attr. $a$

$V_a = \text{mean}(X, \texttt{axis} = \text{`players'})$
$V_P = \text{mean}(P, \texttt{axis} = \text{`players'})$
$\Delta_a = V_a - V_P$
**Returns** $\Delta_a$

---

