# OpenReview forum: "Generative Modeling of Individual Behavior at Scale"
_ICLR.cc/2025/Conference — Submitted to ICLR 2025_

### Official Review · Reviewer_2MDo · 2024-10-28

**Soundness:** 3
**Presentation:** 3
**Contribution:** 2
**Rating:** 6
**Confidence:** 3

**Summary:**

The paper introduces a way to model individual human behavior in games with a scalable, generative approach to behavioral stylometry. Each player is treated as a unique task in a multi-task learning framework, where the authors use parameter-efficient fine-tuning (PEFT) to create style vectors capturing each player’s playstyle. These vectors can then be used to activate shared “skill” parameters, letting the model generate actions tailored to each player. They apply their method to large datasets from chess and Rocket League, and scale their method to tens of thousands of players.

**Strengths:**

- Their architecture and learning procedure was well motivated and explained; for instance, they tackle the transfer/interference tradeoff in multitask settings by using Polytropon.
- Rather than having to fine-tune a separate model for each person, their approach supports large-scale behavioral modeling by assigning unique style vectors to individuals, which activate specific combinations of shared parameters.
- The model doesn’t just classify or predict; it generates actions in the style of individual players, providing a more dynamic and flexible tool for studying human behavior.
- The methodology is tested in two distinct gaming environments—chess and Rocket League; and the authors applied their model to a substantial dataset, covering tens of thousands of players.

**Weaknesses:**

- The paper’s primary contribution is restricted to stylistic adaptation in gaming, without broader implications for other domains. The methodology demonstrates success in chess and Rocket League but fails to show convincingly how these results would generalize to other forms of human behavior modeling.
- It would improve the paper to see more baselines. For example, the authors state "We do not compare to the Transformer-based embedding method because it is incapable of generating moves," however it can still be a good baseline to compare to.
- The authors compare to r McIlroy-Young et al., however, I am not sure if the same test set is used which could make the comparison not that strong?

**Questions:**

1. How do you see this model generalizing to non-gaming applications?
2. Can you provide practical examples where style steering would be beneficial?
3. Have you considered the case when a player’s behavior changes significantly over time?

---

> ### Author Response · Authors · 2024-11-21
> **Response**
>
> Thank you for your review. Regarding our contributions being limited to stylistic adaptation in gaming, Chess and Rocket League are very different games, despite being within the gaming domain. Additionally, the way we train Rocket League is not so different from how language models are trained (we use GPT-2 [2] as our base model). As a result, we see our methodology generalizing fairly easily to any domain where there is a partitioning of the dataset (e.g., into individual players/people). As a more general application of our methodology to a domain other than games, we have applied our method to an image generation setting using diffusion models, as described in our general response above.
>
> Regarding the comparisons to McIlroy-Young et al, we acquired the splits directly from the authors of the original papers for the 400 players stylometry test. However, we were unable to access the sampling method used to create the larger dataset for their unseen tests. As a compromise, we used the same exact player filtering criteria and the same timeframe to source the larger 10,000 player dataset that we used in our few-shot test. This evaluation is significantly more difficult, as it contains thousands of additional players that are trained independently of each other (recall that the LoRA matrices are frozen during few-shot training) and should be a superset of the original dataset. For this reason, we believe that this comparison is more than fair, as we test on a substantially more difficult task and improve significantly on their results.
>
> We address your questions below.
> 1. Please see above.
>
> 2. We can use steering to change a model’s behavior after training it. This is especially useful when there are behaviors that a user may want to change. For example, in chess, if a player is struggling against players who use their bishops effectively, a model that uses bishops more frequently can be generated without training by adding the corresponding style vector. More generally, if a user desires a different distribution of outputs, our method can be used to drive the model closer to that desired distribution, similarly to what fine-tuning would do. The major difference between steering and fine-tuning, however, is that there are no gradient calculations required after MHR has been trained. This makes the process instantaneous and cheap.
>
> 3. We discuss this in the appendix A.4. Additionally, the players in Figure 3 will have sets of data that are not necessarily from the same timeframe, yet still show high similarity to other vectors from the same player. We additionally train very few-shot vectors in Figure 9 of the appendix, which show that the similarity to a vector trained on all games is still several standard deviations above the mean. A split of 25 games may not contain all variations of a player’s playing style, but still contains enough information to learn the general tendencies of a player, even across time. Indeed, the P-value for the similarity of a vector trained on even just 10 games (roughly 2 standard deviations above population mean) is about 0.0455, which is statistically significant.
>
>
> [1]  Y. Zhang, A. P. Jacob, V. Lai, D. Fried, and D. Ippolito, Human-aligned Chess with a Bit of Search, Arxiv 2024.
>
> [2] A. Radford, J. Wu, R. Child, D. Luan, D. Amodei, and I. Sutskever, Language Models are Unsupervised Multitask Learners, 2019.

---

> ### Comment · Reviewer_2MDo · 2024-11-21
> **Response to Rebuttal**
>
> Thank you for the additional experiments, particularly the application to image generation. This demonstrates potential generalizability! However, I am still concerned that the effectiveness of the method hasn't been completely demonstrated. In particular:
> - While you highlight extensions to image generation, the paper does not compare your method against established baselines such as CLIP-guided editing, DreamBooth, or StarGAN. Demonstrating superiority or complementarity to such methods would significantly strengthen the claims of broader applicability.
> - Regarding the McIlroy-Young et al. evaluation, your defense about dataset construction is reasonable. However, to ensure fairness and comparability, applying their model to your new test set is necessary. Reporting their original scores on a different test set does not provide a direct comparison and risks misinterpretation.
>
> These details about the test set with respect to McIlroy-Young et al. seem rather important, and I'm wondering why these details were not included in the original paper?

---

> > ### Comment · Reviewer_2MDo · 2024-11-21
> > **Response to Rebuttal #2**
> >
> > Also, with respect to my second question in my original response, I was asking more generally beyond gaming. I think the paper would be much stronger if the authors added a paragraph discussing an array of concrete instances their method could be applied to for potential future work. This seems like one of the natural ways to contextualize the generalizability of their approach, but this hasn't been done.
> >
> > If the authors can address these concerns that I've outlined, I'd be likely to raise my score.

---

> ### Author Response · Authors · 2024-11-22
> **Regarding your comments**
>
> Thank you for your response! We address the generalizability point and the comparison to McIlroy-Young et al. below.
>
> The purpose of implementing the image generation example was to show the generalizability of our adapter framework and style steering method to a new domain. Although we do not have enough time during the ICLR discussion period to evaluate all of the alternative methods you mentioned, we believe our approach can complement these methods while enabling finer control. Currently, we are able to compare against DreamBooth using LoRA by leveraging existing online scripts [1]. We find that our method allows for more granular control in the final image than DreamBooth, while also retaining more of the original image. You can view this comparison in the link below; we have also added it to the supplementary PDF and have updated our general response above.
>
> **Link to DreamBooth comparison samples:** https://imgur.com/a/9teIqAk.
>
> Regarding applying our method to domains outside of gaming, we will add discussion of concrete examples to the paper. For example, our methodology can be used in the following applications: in image generation, performing fine-grained editing of an image (as we have demonstrated); in LLMs, mitigating or emphasizing stylistic aspects of the generated text; in racing, predicting how human drivers will respond to changes in the car’s aerodynamics (this is part of our ongoing work).
>
> Regarding the comparison to McIlroy-Young et al. for unseen few-shot players, we note that both their base model and ours were trained on the same exact dataset of 400 players. For testing, we reached out to the authors asking for both the model and the test set, but the authors were unable to provide them. As a result, we performed a test that we believe is strictly more difficult, over a much larger universe of 10K players. Note that McIlroy-Young et al.’s method can only perform stylometry — it cannot create generative models of individual behavior or allow style steering, which is the main goal of our work. In order to directly compare the stylometry numbers, one thing we could do is to recreate McIlroy-Young et al.’s model and evaluate it on our test set of players. We cannot complete this in time for the ICLR discussion period, but we can certainly do so before the camera-ready deadline (and will also reach out again to the authors). We will also update Table 1 and our caption/text description to accurately reflect the comparison we did, so that there is no room for misinterpretation.
>
> If you have any further questions, please feel free to ask.
>
> [1] https://github.com/huggingface/peft/blob/main/examples/lora_dreambooth/train_dreambooth.py

---

> > ### Comment · Reviewer_2MDo · 2024-11-24
> > **Response to Rebuttal #3**
> >
> > The applications to other domains are now clear—thank you for providing those insights and examples. I encourage the authors to include this discussion in the paper, as it adds valuable context. Since the authors addressed my concern about generalizability, I increased my rating by a point.
> >
> > I do want to state that this paper has lack of rigorous evaluation of the method to show that its better than existing and alternative approaches. The paper only compares its method to McIlroy-Young et al., with no additional baselines. Even for this comparison, the test set differs, and while the authors argue their test set is more challenging, this cannot be definitively verified. If additional baselines were included, this issue would not be as problematic, but the authors rely solely on this one comparison. Additionally, it is kind of concerning that the details about McIlroy-Young et al.’s test set differences were not outlined in the original submission.
> >
> > To conclude, if the authors had more thoroughly tested their method against other baselines, indeed this paper would more definitively be a strong contribution to the field.

---

### Official Review · Reviewer_MvwK · 2024-11-02

**Soundness:** 2
**Presentation:** 3
**Contribution:** 4
**Rating:** 10
**Confidence:** 3

**Summary:**

The main focus of this paper is behavioural stylometry---how can one efficiently build AI agents that are customized to individual users.
The proposed method involves simultaneously finetuning several LoRA adapters as well as a set of mixing rates indicating how much each LoRA adapter should contribute to the prediction for any particular user. The paper uses two games as a case study to evaluate their approach: chess and a soccer-like video game. In both cases, they show their method effectively predicts user moves. The paper further explores how a system trained on some number of users can be easily extended in a few-shot setting to new users.

**Strengths:**

This paper was a fascinating read, and both the methodology and the experiments used to support its efficacy are quite compelling. While the method being employed (`Poly` with multi-head routing) was already introduced in prior work, this prior work focused on a handful of NLP tasks rather than behavioural modeling for style-customized agents.

After reading the paper, I was left interested in trying out the MHR finetuning approach in other problem domains I'm interested in, and I feel fairly confident in my ability to reproduce the proposed method using the details in the paper.

**Weaknesses:**

1. I am slightly concerned that, due to me being non-expert in this area, my perception of the proposed method's novelty is greater than it actually is. I would like to see additional explanation in the "Background and Framing" section situating the paper's contributions relative to those in the Polytropon and MHR papers.
2. The majority of the experiments are only on the chess domain. I wold have liked to see more reproduction of experiments on the Rocket League domain.
3. I would have liked to see a user study where human players are asked to assess the style of different agents.
4. I would like to see more analysis of how few-shot performance for unseen players varies as a function of the amount of data available for tuning the new player's style vector. E.g. a figure plotting move prediction accuracy as a function of number of games used for tuning.

**Questions:**

1. What do the bold numbers mean in Table 1?
3. I don't understand what the y-axis is in Figure 7.

**Details Of Ethics Concerns:**

The authors develop a method which can be used to (1) de-identify anonymous user behaviour data, and (2) create agents that mimic the behaviour style of individual users. They evaluate their methods on two extremely benign game domains. However, since methods for impersonation and de-identification of users can have negative consequences in other domains, I have opted to flag this paper.

---

> ### Author Response · Authors · 2024-11-21
> **Response**
>
> Thank you for your review. Indeed, the potential of applying MHR fine-tuning and style steering to other domains is an exciting prospect!
>
> As you observed, we do not claim novelty over the PEFT techniques (Polytropon, MHR) used in this paper, which are typically used to train low-rank adapters for multi-task learning in NLP. Instead, we believe our novelty comes from our application and adaptation of a multi-task adapter framework to behavioral modeling and stylistic analysis. We find that these adapters create an interpretable and steerable style space, which we validate through chess and Rocket League, as well as the image generation example shown in our general response. Our style methodology and style delta vector method in particular enable new capabilities, such as the ability to create new styles and steer (change) existing styles in a human-interpretable way.
>
> Regarding a user study, we agree that this would be a very interesting and relevant investigation. In our paper, we have reported a quantitative set of heuristics (that were designed by chess masters and software engineers) to characterize playing style, though it would be interesting to see how humans perceive these changes. We have consulted with chess masters in the past to validate the effect of steering a player’s style vector, comparing moves made by the player’s model before and after the change. However, we have found that the quantitative heuristics allow us to validate the style change more effectively at scale.
>
> We do not provide such a steering analysis for Rocket League due to computational constraints in the environment rollouts, and a lack of single-step heuristics from data points, as many of the existing heuristics depend on behaviors over longer trajectories. It is difficult to generate a reliable and consistent benchmark for this game as it would require hundreds of thousands of rollouts per player in a simulator that can only handle on the order of hundreds of frames per second. As a more general application of our methodology to a domain other than games, we have applied our method to an image generation setting using diffusion models, as described in our general response above.
>
> Regarding few-shot performance, please refer to Figure 2, where we show move-matching performance relative to the number of games available when training a model.
>
> Regarding your questions,
> 1. The bold numbers show which of the methods perform better in that sub-task. For all of the shown tasks, we outperform previous work.
> 2. The y-axis in figure 7 is the difference (in standard deviation) between the model’s behavior before and after modifying the model parameters. For example, for the “bishop pair” numbers, we first note the behavior of the model before the modification. We then standardize this number over all players ((feature - feature_mean) / feature_std). We then apply the bishop delta vector by adding to the parameters of the model. We then roll the model out on the same set of positions and see what changed. After, we standardize with the original model’s std and mean ((feature_changed - feature_mean) / feature_std) and subtract from the original numbers. This delta is the change in the model behavior after a change in the model parameters.
>
> We hope the responses above address your concerns. Please let us know if you have any further questions.

---

> > ### Comment · Reviewer_MvwK · 2024-11-22
> > **Response**
> >
> > Thanks for your response. I strongly encourage you to revise your draft to make these points a bit clearer.
> >
> > Also, in general, I think all your figures/tables could benefit from longer captions that briefly explain the insight the reader is supposed to garner from the figure/table.

---

> > > ### Author Response · Authors · 2024-11-25
> > >
> > > Thank you, reviewer MvwK! We will revise our draft as you suggested and appreciate your thoughts and comments throughout this review process.

---

### Official Review · Reviewer_67Ho · 2024-11-04

**Soundness:** 2
**Presentation:** 2
**Contribution:** 3
**Rating:** 5
**Confidence:** 4

**Summary:**

The paper is focusing on modeling individual human behavior in games. Unlike previous works which mainly focus on modeling human behavior at an aggregate level, the authors propose learning an individualized approach for generating actions in the style of each player. To this end, they use a behavioral cloning model based on a multi-task learning approach combined with parameter-efficient finetuning (PEFT) to learn a shared set of skills across players, as well as style vectors that induce a generative model for each player. They use this approach for style steering new players towards desired properties. The authors experiment on two games, chess and RocketLeague, and find that their approach is comparable to the performance of behavioral cloning methods. They also demonstrate they can manipulate the behaviour of players and steer them towards human-interpretable attributes such as, for example, interpolating between player characteristics and style steering from a weaker player’s style to a stronger player’s style.

**Strengths:**

The multi-task method for modeling players proposed in this paper presents the advantage of being scalable to many players, it is parameter-efficient and allows for human-interpretable control of player attributes

The authors conduct experiments on two games, chess and RocketLeague, and find the proposed method achieves performance comparable to behavioral cloning methods that do full finetuning of the model parameters

Style vectors encode different player skills, and can be combined, interpolated, and steered towards desirable human-interpretable attributes to change the playing style of each player; the analysis of these vectors reveals they are consistent within a single player and across different players.

Although the approach combines existing methods in the literature, it applies them in a novel context (player modeling and human-interpretable player steering). The synthesis of new styles experiments show that it is possible to employ basic arithmetic on style vectors to interpolate between players of different strengths and skill levels and steer player styles in desirable directions.

**Weaknesses:**

While the proposed method is parameter efficient and needs only few shot examples for each player, the performance is often lagging behind state-of-the-art behavioral cloning methods (Figure 2)

Unclear if results in Table 1 are statistically significant, if proposed method surpasses previous approaches and the results are reported across same set of players

The paper needs additional clarifications and details (please see comments and questions below):


Additional comments:

In the unclear in the introduction in which contexts, tasks and domains this approach is relevant. Only towards the end of the introduction it is mentioned this approach is used in game environments for modeling players.

Line 89 - the authors claim they introduce the notion of style vectors, whereas this is already well established in the literature, particularly in NLP

Line 96 - it would be desirable to briefly summarize the insights of these analyses

Figure 1 - it is unclear which are the MHR adapters and which is the routing matrix in the figure

Lines 230-231 - missing citation for the original Maia model

Line 262 - move-matching accuracy is introduced without explaining what this metric represents and how it is computed

Lines 273- 274 - “our results can be interpreted in this way” - please detail

Line 358: I would suggest replacing “universe” with “environment” (same in Table 1)

**Questions:**

How many reference set of games are available for each player during few-shot learning?

Table 1 - results for McIlroy-Young et al. (2022b) and McIlroy-Young et al. (2021) are borrowed from their respective papers; how do we know these are the same set of players to make the comparison fair?

Why for the unseen few-shot players you are only comparing to McIlroy-Young et al. (2021) ?

What does Random (%) denote and why are all results in that column 0.25?

Figure 2 - as the game count increases, the performance of MHR-MAIA decreases. How do you explain that? More analysis should be provided in the paper (Section 5.1) of why this happens; Section 5.2 does include some details, however they are coming in late for the reader.

---

> ### Author Response · Authors · 2024-11-21
> **Response**
>
> Thank you for your detailed feedback. We have integrated your comments into the submission. Regarding lagging performance, in Figure 2, we observe that our method performs very similarly to state-of-the-art behavior cloning methods, while being around 2 orders of magnitude more efficient (see lines 365-369 of our paper). The difference in move-matching accuracy is very minimal across all game counts, and within the margin of error. In other words, we are much more efficient for similar performance, which is crucial for scaling.
>
> We provide answers to your questions below.
>
> 1. For Table 1, there are 100 games for few-shot learning for the larger player sets. We test with as few as 25 games in other charts, such as in Figure 2 and Figure 9.
>
> 2. For the 400 players stylometry test, we acquired the splits directly from the authors of the original papers. However, we were unable to access the sampling method used to create the larger dataset for their unseen tests. As a compromise, we used the same exact player filtering criteria and the same timeframe to source the larger 10,000 player dataset that we used in our few-shot test. This evaluation is significantly more difficult, as it contains thousands of additional players that are trained independently of each other (recall that the LoRA matrices are frozen during few-shot training) and should be a superset of the original dataset. For this reason, we believe that this comparison is more than fair, as we test on a substantially more difficult task and improve significantly on their results.
>
> 3. Fully fine-tuned models of individual behavior (McIlroy-Young 2022b) are too computationally expensive to train at scale (please see lines 364-367 in Section 5.1 of our updated submission). That paper does not report any results in the unseen few-shot setting on a larger player set. McIlroy-Young et al. (2021) creates a player embedding space, which is more scalable in the stylometry setting, but does not yield individual models of player behavior.
>
> 4. Random (%) represents the naive baseline of selecting a player at random for the stylometry task. It is supposed to be a measure of how difficult that stylometry task is relative to the existing set of players.
>
> 5. The performance of MHR-Maia decreases relative to the individual models by a very small margin, especially considering that there are relatively few players with over 10,000 games across the Lichess platform. One possible explanation behind this difference is that MHR-Maia only has ~256 parameters to store all the skills for an individual player when fine-tuning towards them. This is in comparison to the full model which has several million. We will move the performance discussion earlier into the paper where Figure 2 is introduced. We predict that increasing the rank and skill count of the MHR adapters will bring us in parity or exceed individual fine-tuning, but that may reduce the interpretability of this space. This line of analysis (dimensionality vs. interpretability) would be an interesting line of future research.
>
> Please let us know if you have any further questions or comments.

---

> ### Author Response · Authors · 2024-11-26
>
> Dear reviewer 67Ho,
>
> We hope you have had a chance to review our responses to your individual concerns, as well as our general response above regarding the generalizability of our approach. We have added a new experiment in a new domain (image generation) that should address many of your concerns.
>
> If you have any further questions we can answer to help you revise your assessment of our paper, please let us know. Your consideration of our responses and the discussion we've had with other reviewers is much appreciated.
>
> Thank you!

---

### Official Review · Reviewer_Ma2G · 2024-11-04

**Soundness:** 2
**Presentation:** 2
**Contribution:** 2
**Rating:** 5
**Confidence:** 3

**Summary:**

The paper studies the modeling of individual behaviors. The learning problem is modeled to multi-task learning. Experiments are made on two large scale games, chess and Rocket League.

**Strengths:**

1. The paper proposes a novel method to model the individual behavior (player behavior) by applying PEFT (specially LoRA), learning a style vector for each player. In addition, the style vectors allow for the generation of actions by steering. These ideas are novel.

2. Experiments are carefully-designed and confirm the representativeness of the learned style vectors.

**Weaknesses:**

1. The abstract is confusing. I am not an expert to the concerning domain. For example, what is the purpose to model human behavior using AI? Does the research only contribute to games, or some other domains?

2. Keeping up with the last point, the proposed methods can too specific in the chosen domain. It is not clear whether the proposed method can contribute to a broad range of audience, or contributing to the AI community.

3. I have some questions about the experiment settings. What is the base model used in the experiments?

4. Can the authors explain more on the usage of strong LLMs (e.g. GPT-4o) on the concerning task. For example, if one logging the specific player behavior in the context of GPT-4o, can it perform well in simulating the player?

**Questions:**

Please see above.

---

> ### Author Response · Authors · 2024-11-21
> **Response**
>
> Thank you for your review. We address each question below.
>
> 1. As mentioned in our introduction, modeling human behavior in games has many potential applications, including identifying weaknesses in a human’s play and developing personalized training material, developing better AI partners or teammates [5], and creating more realistic or enjoyable playing experiences [6]. The Maia chess bots on Lichess [6] are extremely popular: they have played millions of games with humans (orders of magnitude more than any other bots). However, human behavioral modeling has also been used in other domains, such as in economics to automate supply chain decision making [1], Formula 1 racing to predict how human drivers will respond to changes in the car’s aerodynamics*, and legal cases to provide more reliable models of judges’ decisions—showing that it has broader uses in other fields.
>
> 2. The proposed methods in our paper show that a black-box adapter framework inspired by multi-task learning in NLP can be applied to a domain like gaming, allowing us to learn individualized models of player behavior. In addition, our method learns consistent features about each player that can be interpreted and steered in a black-box manner. As far as we know, this is the first type of work that is capable of doing this. We also show a more general application of our method to steer the output of a diffusion model in a fine-grained manner (see general response above).
>
> 3. The base models along with their exact dimensionalities and parameter counts are detailed in Section 3.1 of the paper. For chess, we use a Squeeze and Excitation Residual Network (SEResNet) [2]. For Rocket League, we use GPT-2 [3] initialized and trained from scratch with a linear output map to the action space at each encoded state.
>
> 4. We unfortunately do not have a baseline with GPT-4o. Prompting an LLM with historical games from an individual player to generate each move in a game would be prohibitively expensive. Nevertheless, a recent paper [4] provides a similar (non-individualized) baseline using GPT 3.5, and shows 53.7% accuracy. They do not fine-tune the model to individual players, though the accuracy they report is similar to other base models before fine-tuning. Their work also suggests that this domain is very data limited, rather than parameter limited.
>
> We hope this was helpful. Please let us know if you have any further questions.
>
> [1] D. Kurian, V. Pillai, J. Gautham, and A. Raut, Data-driven imitation learning-based approach for order size determination in supply chains. EJIE 2023.
>
> [2] J. Hu, L. Shen, S. Albanie, G. Sun, and E. Wu, Squeeze-and-Excitation Networks. CVPR 2019.
>
> [3] A. Radford, J. Wu, R. Child, D. Luan, D. Amodei, and I. Sutskever, Language Models are Unsupervised Multitask Learners, 2019.
>
> [4] Y. Zhang, A. P. Jacob, V. Lai, D. Fried, and D. Ippolito, Human-aligned Chess with a Bit of Search, Arxiv 2024.
>
> [5] K. Hamade, R. McIlroy-Young, S. Sen, J. Kleinberg, and A. Anderson, Designing Skill-Compatible AI: Methodologies and Frameworks in Chess. Arxiv, 2024.
>
> [6] Lichess bots page. Lichess, 2024. Available: https://lichess.org/player/bots
>
> *Ongoing work by authors, blinded for anonymity

---

> ### Comment · Reviewer_MvwK · 2024-11-22
> **On Weakness #2**
>
> I would also like to push back against point 2, that only showing the method works for two game domains is a weakness of the paper.
>
> Game-playing agents are perhaps the most classical application of AI, with research dating to the 1960s on building AI for chess. This is the first paper I've seen to develop a chesbot that tries to efficiently model and replicate individual user behaviour. Even if the proposed method didn't have other applicability in other domains (which I don't think is true), this contribution is worth sharing with the community.

---

> ### Author Response · Authors · 2024-11-26
>
> Dear reviewer Ma2G,
>
> We hope you have had a chance to review our responses to your individual concerns, as well as our general response above regarding the generalizability of our approach. We have added a new experiment in a new domain (image generation) that should address many of your concerns.
>
> If you have any further questions we can answer to help you revise your assessment of our paper, please let us know. Your consideration of our responses and the discussion we've had with other reviewers is much appreciated.
>
> Thank you!

---

### Author Response · Authors · 2024-11-21
**General Response**

We thank all the reviewers for their thoughtful reviews, and have responded to each review below. Several reviewers expressed concerns about the generalizability of our method; to address this, we have applied our MHR training method and style delta vector steering method to a new domain: image generation. Specifically, we took a Stable Diffusion 1.5 [2] base model and applied MHR fine-tuning using images of celebrities from the CelebA Faces with Attributes dataset [1], allowing us to learn a style vector for each of the 10,177 celebrities in the dataset. Then, we computed style delta vectors for certain attributes and used them to steer the output of the model. We used an image resolution of 256x256.

Below, we compute “No Beard”, “Smiling”, and “Black Hair” style delta vectors using cosine similarities between the CLIP [3] image and text embeddings for the images and features provided in CelebA. The style delta vector computation is identical to the method described in Section 4 of our paper. We provide a series of images generated using these vectors, starting with the un-steered image, followed by an image generated with the style delta vector, and finally an image generated with a higher weighting of the style delta vector. The same initialization noise is used to generate each of the three images. We compare our method against DreamBooth at the request of one of the reviewers, and find that our method is able to achieve more fine-grained control without significantly changing the base image, unlike DreamBooth.

We find that our method is able to generalize to this non-gaming application that is of interest to the ML community (editing generated images in a granular manner), even though our paper’s focus is on human behavioral modeling. We have included this result Figure 8 of the appendix, and mention it in the main paper.

**Link to steered samples:** https://imgur.com/a/9teIqAk

The samples are additionally available in Figure 8 of the revised submission and supplementary material.

[1] Z. Liu, P. Luo, X. Wang, and X. Tang, Deep Learning Face Attributes in the Wild, ICCV 2015.

[2] R. Rombach, A. Blattmann, D. Lorenz, P. Esser, and B. Ommer, High-Resolution Image Synthesis With Latent Diffusion Models, CVPR 2022.

[3] A. Radford, J.W. Kim, C. Hallacy, A. Ramesh, G. Goh, S. Agarwal, G. Sastry, A. Askell, P. Mishkin, J. Clark, G. Krueger, I. Sutskever, Learning Transferable Visual Models From Natural Language Supervision, Arxiv 2021.

---

### Author Response · Authors · 2024-12-04
**Final note and thanks**

We thank the reviewers for their valuable feedback and comments. We have made changes to the paper reflecting these, and have added an additional application to show the generalizability of our method. Although we have not heard from Reviewers Ma2G and 67Ho, we hope that they reconsider their assessment of our paper in light of the new generalization experiment, edits, and discussion with reviewers MvwK and 2MDo. Thank you!

---

### Meta-Review · Area_Chair_fJsS · 2024-12-19

**Metareview:**

The authors consider the task of imitating / generating behaviors of individuals (stylometry). They use PEFT style methods to obtain generative models that are matched to specific individuals and then identify shared components to be able to do style steering. Evaluations are done on chess (esp comparisons to Mcilroy-young) and rocket league, and on tasks of predicting moves, as well as style steering.

 The reviews are variable, and the strengths and weaknesses reflect the perspectives of the reviewers - on the one hand, there are interesting ideas here involving the use of PEFT for stylometry, and the potential of identifying high-level control mechanisms for style. On the other hand, there are quite questionable parts of the paper, including issues highlighted by 2MDo and others that the comparison with McIlroy-young are not exactly comparable (as the numbers are not matched, re-run baselines but rather numbers copied from the original paper where the settings differ slightly).

I see both these arguments, though I lean on the side of thinking that these issues involving baseline comparisons are problematic - it's one thing to have weak comparisons, but in this case, the baseline numbers themselves are likely not something we should be comparing to at all.

**Additional Comments On Reviewer Discussion:**

Reviewers 2MDo and MvwK engaged in discussions both with the authors and the AC in the post-author discussion period. Most of the points raised by the reviewers were somewhat subjective (e.g. the broader audience point of Ma2G), but 2MDo discovered during the discussion back and forth some of the issues with the baseline comparisons. The authors have promised to update some of these comparisons by camera-ready, but that does not seem like a reasonable mitigation for a fairly serious issue.

---

### Decision · Program_Chairs · 2025-01-22

Reject